# Hyaluronan and Collagen Are Prominent Extracellular Matrix Components in Bovine and Porcine Ovaries

**DOI:** 10.3390/genes12081186

**Published:** 2021-07-30

**Authors:** Wendena S. Parkes, Farners Amargant, Luhan T. Zhou, Cecilia E. Villanueva, Francesca E. Duncan, Michele T. Pritchard

**Affiliations:** 1Department of Pharmacology, Toxicology and Therapeutics, University of Kansas Medical Center, Kansas City, KS 66160, USA; wparkes@kumc.edu (W.S.P.); cvillanueva@kumc.edu (C.E.V.); 2Department of Obstetrics and Gynecology, Feinberg School of Medicine, Northwestern University, Chicago, IL 60611, USA; farners.amargant@northwestern.edu (F.A.); tracy.zhou@northwestern.edu (L.T.Z.); 3Institute for Reproductive and Perinatal Research, University of Kansas Medical Center, Kansas City, KS 66160, USA

**Keywords:** stroma, extracellular matrix, hyaluronan, hyaluronan synthase, hyaluronidase, collagen, ovary, bovine, porcine

## Abstract

The extracellular matrix (ECM) is a major component of the ovarian stroma. Collagen and hyaluronan (HA) are critical ovarian stromal ECM molecules that undergo age-dependent changes in the mouse and human. How these matrix components are regulated and organized in other mammalian species with reproductive characteristics similar to women such as cows and pigs, has not been systematically investigated. Therefore, we performed histological, molecular, and biochemical analyses to characterize collagen and HA in these animals. Bovine ovaries had more collagen than porcine ovaries when assessed biochemically, and this was associated with species-specific differences in collagen gene transcripts: *Col3a1* was predominant in cow ovaries while *Col1a1* was predominant in pig ovaries. We also observed more HA in the porcine vs. bovine ovary. HA was distributed across three molecular weight ranges (<100 kDa, 100–300 kDa, and >300 kDa) in ovarian tissue and follicular fluid, with tissue having more >300 kDa HA than the other two ranges. Transcripts for HA synthesis and degradation enzymes, *Has3* and *Hyal2*, respectively, were predominant in cow ovaries, whereas *Has2*, *Kiaa1199,* and *Tmem2* tended to be predominant in pig ovaries. Together, our findings have implications for the composition, organization, and regulation of the ovarian ECM in large mammalian species, including humans.

## 1. Introduction

The mammalian ovary is a highly dynamic structure that undergoes remarkable hormone-mediated functional and structural changes which promote follicle development, ovulation, and subsequent ovarian wound repair throughout its reproductive lifespan [1,2,3,4]. Follicles are the functional units of the ovary which develop in a complex microenvironment comprised of extracellular matrix (ECM) molecules, smooth muscle cells, fibroblasts, immune cells, endothelial cells, and neurons [5]. The ECM is made up of various components such as collagen, glycoproteins, and proteoglycans [6]. The ECM’s role includes providing structural support and aiding in cell signaling and migration [7]. Two ubiquitous extracellular matrix molecules that have an important role in ovarian biology are collagen and hyaluronan (HA). There are 28 different members of the collagen family that can also undergo post-translational modifications [8]. We found that with increased reproductive age, there is an accumulation of type I and III collagen resulting in ovarian fibrosis in mice [3,9,10]. Collagen accumulation with reproductive age also occurs in women [9]. Interestingly, enzyme-mediated collagen depletion of old mouse ovaries, ex vivo, decreases their stiffness and restores their biomechanical properties to those of young ovaries, demonstrating a major functional role of ovarian collagen [9]. The presence of mechanical stress retains primordial follicle dormancy, and this is associated with an abundance of collagen around the follicles [11]. Moreover, secondary follicle growth in vitro is restricted if follicles are grown in higher percentage alginate hydrogels than control follicles [12], suggesting that increased stiffness negatively affects follicle growth. 

Similar to collagen, HA also plays a role in regulating ovarian biomechanics and tissue homeostasis [9]. HA is a glycosaminoglycan made of repeating N-acetyl-glucosamine and D-glucuronic acid disaccharide units [13]. HA is a high molecular weight (HMW) molecule synthesized at the plasma membrane by HA synthase enzymes (Has1, Has2, Has3) [14], where it can remain attached to the cell as part of the glycocalyx or be released and incorporated into the ECM. These three HA synthases share 59–71% identity [15]. HMW HA is homeostatic and helps to provide tissue hydration and lubrication as well as cell quiescence and survival signals [16,17]. However, HA can be fragmented by reactive oxygen species (ROS) or hyaluronidases to low molecular weight (LMW) HA, which creates a reactive matrix resulting in inflammation [18]. This inflammation can cause additional tissue damage but is also required for tissue repair by promoting cell proliferation, migration, and angiogenesis [17,19]. Hyaluronidases include Hyal1, Hyal2, Tmem2, and Kiaa1199. Although Hyal1, Hyal2, and Tmem2 share considerable homology at the DNA level, Kiaa1199 is unique because it does not have homology to the other hyaluronidases [20]. Our laboratories have previously shown that ovarian stromal HA is lost with advanced reproductive age in mice, and these age-dependent changes are conserved in women [3,9,10]. In Has3 KO mice, which have a mutation in the Has3 catalytic domain leading to a loss of enzymatic function, there is a trend towards decreased HA in comparison to WT mice, and this is associated with increased tissue stiffness [9]. In support of a direct role for HA fragments in driving age-associated inflammation in the ovary, treating ovarian stromal cells with LMW HA induces a pro-inflammatory gene expression program [21]. Collectively, these data suggest that collagen and HA are important for ovarian homeostasis, possibly through signaling events or tissue micromechanics.

Given the importance and prevalence of collagen and HA in the mouse and human ovary, there is a need to characterize these ovarian matrix components in models beyond the mouse, especially in large mammalian species such as cows and pigs. These species are important models as they have similar reproductive parameters to women, including the length of the ovulatory cycle follicular phase, luteal phase, and duration of gestation [22,23]. Although some published research has explored bovine and porcine ovarian stroma, a comprehensive analysis of collagen and HA matrices in cows and pigs in parallel has not been completed [24,25,26,27,28,29,30]. Therefore, in this study, we used qualitative and quantitative analyses to define ovarian collagen and HA content and the network of genes regulating these two ECM molecules in both species. Thus, this work broadens our understanding of stromal biology using two additional mammalian species and provides a foundation for further exploration of the ovarian microenvironment in women and in critical agricultural species.

## 2. Materials and Methods

### 2.1. Ovarian Tissue and Follicular Fluid Collection

Reproductively young CB6F1 mice (3–4 weeks old) were obtained from Envigo (Indianapolis, Indiana). These mice were housed at the University of Kansas Medical Center (KUMC) under constant temperature, humidity, and light (10 h light/14 h dark cycle) with free access to food and water. They were aged to 6 weeks old before use. All animal experiments described here were approved by the KUMC Institutional Animal Care and Use Committee and were performed in accordance with National Institutes of Health Guidelines. Another set of reproductively young CB6F1 mice (6–12 weeks old, Envigo) were housed at Northwestern University’s (NU) Center for Comparative Medicine under constant temperature, humidity, and light (14 h light/10 h dark) with free access to food and water. They were used at 6–12 weeks old. All animal experiments described here were approved by the NU Institutional Animal Care and Use Committee and were performed in accordance with National Institutes of Health Guidelines.

Bovine (*Bos taurus*) ovaries were obtained from the Aurora Packing Company (Aurora, IL, USA) and transported by courier service to NU within two hours of harvest. The Aurora Packing Company is an abattoir that processes cattle from multiple sites across the Midwest within a 300-mile radius of Chicago. While unable to control for the specific breed from which the ovaries were harvested, the company primarily processes beef cattle (Angus breed), and the heifers were post-pubertal and reproductive adult (between 24 and 30 months). We excluded ovaries that had prominent corpora lutea during the tissue collection process.

Follicular fluid was aspirated from all visible small antral follicles (3–6 mm in diameter) from intact and non-ruptured isolated bovine ovaries. The follicular puncture was performed using an 18-gauge disposable needle attached to a syringe. Once all follicles were aspirated, fluid from the syringe was transferred to a conical-ended centrifuge tube, and cellular/tissue debris was allowed to settle (1× *g*) at room temperature for 15–20 min. Afterward, the supernatant was transferred to microtubes and spun down at 1000× *g* for 2 min. After centrifugation, the clarified follicular fluid was transferred to new sterile microtubes, frozen on dry ice, and stored at −80 °C. To prepare tissue enriched for bovine stroma, cumulus-oocyte complexes (COCs) were first isolated from whole ovaries by cutting small antral follicles (3–6 mm) with a scalpel directly into media. After all the COCs were isolated, the remaining ovarian tissue now enriched for stroma was cut into pieces and frozen on dry ice. Prior to RNA isolation, 350 mg of bovine stromal tissue was submerged in RNA later ice (pre-chilled to −80 °C; Thermo Fisher Scientific, Waltham, MO, USA) to stabilize RNA overnight at −20 °C.

Two pigs (*Sus scrofa domesticus*), ages 8 and 9 months, were purchased from Sinclair Bio-resources and were delivered to the University of Kansas Medical Center Laboratory Animal Resources (LAR) facility, where they were housed in metal pens with straw bedding. The animals were provided food, water, and social enrichment ad libitum. Immediately after harvesting, the ovaries were transported to the lab on ice in Leibovitz’s L-15 medium (Gibco, Grand Island, NY, USA) supplemented with 3 mg/mL polyvinylpyrrolidone (Millipore-Sigma, St Louis, MO, USA) and 0.5% penicillin and streptomycin (Thermo Fisher Scientific, Waltham, MO, USA, L15 PVP). Upon arrival, porcine follicular fluid was collected using a 26-gauge disposable needle and 1 ml syringe and centrifuged at 10,000× *g* for 1.5 min at 4 °C to remove cellular/tissue debris, and then the supernatant was stored at −20 °C. The porcine ovaries were dissected into quarters, and one quarter was stored in Modified Davidson’s solution (MDS; Electron Microscopy Sciences, Hatfield, PA, USA), one in Carnoy’s solution (Fisher Scientific, Waltham, MA, USA), one in RNA later (Fisher Scientific, Waltham, MA, USA) and the final was snap-frozen in liquid nitrogen. Porcine ovaries fixed in MDS and Carnoy’s solution were processed and embedded in paraffin for sectioning. Ovaries in RNA were later incubated at 4 °C overnight and then transferred to −80 °C until use. Snap frozen ovaries were stored at −80 °C until use.

### 2.2. Histochemical Staining

To analyze tissue architecture, tissue sections were stained with hematoxylin and eosin (H&E). Samples were deparaffinized in CitriSolv (Decon Laboratories, Prussia, PA, USA) and rehydrated in a series of graded ethanol baths (100%, 75%, 70%), and a standard H&E staining protocol was used [6,7]. Whole tissue scans and single images were taken using an EVOS FL Auto Cell Imaging system as well as and a Nikon Eclipse E600 microscope (single images).

For collagen assessment, picrosirius red staining (PSR) was performed on bovine (*n* = 2 cows, *n* = 1 ovary/cow, *n* = 2 sections/ovary) and porcine (*n* = 2 pigs, *n* = 2 ovaries/pig, *n* = 2 sections/ovary) ovarian histologic sections as previously described [6]. In brief, histologic samples were deparaffinized using CitriSolv and then rehydrated in a series of graded ethanol washes (100%, 70%, 30%). Slides were then incubated in a PSR staining solution, which contains Sirius Red F3D (Direct Red 80, C.I. 35782, Sigma-Aldrich, St. Louis, MO, USA) in a 1.3% saturated aqueous picric acid solution (Sigma-Aldrich, St. Louis, MO, USA) at 0.1% *w/v* for 40 min at room temperature (RT). Then, slides were destained with acidified water for 1.5 min. Finally, slides were dehydrated in 100% ethanol, cleared with CitriSolv, and mounted with Cytoseal (Thermo Scientific, Waltham, MA, USA). Stained slides were imaged in brightfield using an EVOS FL Auto Cell Imaging System (tissue scan and single images) and a Nikon Eclipse E600 microscope (single images).

Masson’s Trichrome (MTC) Assay (Polysciences, Inc., Warrington, PA, USA) was performed on bovine (*n* = 2 cows, *n* = 1 ovary/cow, *n* = 2 sections/ovary) and porcine (*n* = 2 pigs, *n* = 2 ovaries/pig, *n* = 2 sections/ovary) histological samples to visualize collagen using a PSR alternative method. Ovarian tissue sections were deparaffinized using CitriSolv and rehydrated in 2 ethanol baths of 100% and 95% each, followed by immersion in pre-heated Bouin’s solution at 60 °C for 1 h. After a gentle wash in running tap water, the samples were stained in Weigert’s Iron Hematoxylin for 10 min. Samples were washed again with tap water and then incubated with Biebrich Scarlet—Acid Fuchsin solution for 5 min followed by a rinse in distilled water. Next, samples were stained with phosphotungstic/phosphomolybdic acid for 10 min immediately followed by incubation for 5 min with Aniline Blue. After rinsing the slides with distilled water, they were transferred to 1% acetic acid solution for 1 min and then dehydrated with ethanol, cleared with CitriSolv, and finally mounted in Vectashield hardset antifade mounting medium (Vector Laboratories, Burlingame, CA, USA). MTC stained slides were visualized using an EVOS FL Auto Cell Imaging System (tissue scan and single images) and a Nikon Eclipse E600 microscope (single images).

### 2.3. Hydroxyproline Assay

To quantitatively estimate collagen content in ovarian tissue, the hydroxyproline assay was used as previously described [6]. Briefly, 10 mg of tissue from cow, pig, and mouse ovaries were used for this assay. The ovarian tissue was hydrolyzed in 100 µL of 12.1 N hydrochloric acid, and 100 µL of water then incubated for 3 h at 120 °C. The samples were vortexed every 30 min during the incubation period to ensure complete tissue lysis. After incubation, the samples were centrifuged for 10 min at 10,000× *g*. The supernatants for the cow, pig, and mouse samples were added to a 96 well plate by adding 0.5, 1, 2, or 5 µL of sample per well, in duplicate, to which 100 µL of Chloramine T reagent (Millipore-Sigma, St Louis, MO, USA) was added and allowed to incubate for 25 min at room temperature. After incubation, 100 µL of Ehrlich’s Solution (1M 4-(dimethylamino)benzaldehyde in 1-propanol/60% perchloric acid (3:1, *v*/*v*); Millipore-Sigma) was added and incubated at 60 °C for 35 min in a non-humidified, laboratory oven. After the incubation, the absorbance of the samples was measured at 550 nm using a microplate reader (Synergy 2^TM^ Multi-Mode microplate reader, Biotek, Winooski, VT, USA). A standard curve was created using known concentrations from which hydroxyproline content in each sample was calculated. Collagen contains approximately 12.5% hydroxylated prolines [28]; therefore, the collagen found per mg of ovarian tissue was extrapolated by dividing the hydroxyproline concentration by 12.5%.

### 2.4. Real-Time Quantitative Polymerase Chain Reaction (RT-PCR)

Approximately 35 mg of bovine and porcine ovarian tissue samples were added to bead homogenization tubes (Lysing Matrix D tubes MP Biomedicals, Solon, OH, USA) in RLT lysis buffer (RNeasy Mini Kit, Qiagen, Germantown, MD, USA) containing 2-mercaptoethanol (1:100) and homogenized in a FastPrep 24 (MP Biomedicals). The RNA was isolated using the RNeasy Mini Kit, and genomic DNA was digested using on-column DNaseI treatment, according to the manufacturer’s instructions (use of RNA instead of cDNA in the RT-PCR reaction does not generate an amplicon confirming lack of genomic DNA contamination). The RNA was eluted in 30 µL of RNase, DNase-free water, and the quantity and purity of the RNA were determined with a NanoPhotometer (Implen, Westlake Village, CA, USA). RNA was reversed transcribed to cDNA using a High-Capacity Reverse Transcription Kit (Thermo Fisher Scientific, Waltham, MO, USA). RT-PCR was performed to measure HA synthase (*Has1*, *Has2*, *Has3*) and hyaluronidase (*Hyal1*, *Hyal2*, *Tmem2,* and *Kiaa1199*) and *Gusb* (as a housekeeping transcript) cDNA from bovine, porcine, and mouse ovarian tissues using primer–probe assays or SYBR green chemistry as described in Appendix A. To assess ovarian collagen transcripts, we measured *Col1a1* and *Col3a1* in bovine ovarian stromal tissue, porcine ovaries, and mouse ovaries and again used *Gusb* as a housekeeping transcript. For this analysis, we only used SYBR green chemistry. Primer sequences for cow and pig, where available, are found in Appendix A. For mouse, primer sequences were retrieved from the PrimerBank (https://pga.mgh.harvard.edu/primerbank/, accessed several times from July 2011–July 2017), were designed by a colleague, or were retrieved from the peer-reviewed literature; all of these primers were synthesized by Integrated DNA Technologies (Coralville, IA, USA) [29]. All reactions and thermal cycling conditions were performed according to the manufacturer’s instructions and were executed in a CFX384 Touch Real Time PCR System (Bio-Rad, Hercules, CA, USA). The thermal cycling conditions were as follows. For cow and mouse, *Has, Hyal,* and *Gusb* (Taqman Primer Probe Sets, Invitrogen, Pleasanton, CA, USA,) reactions were incubated at 50 °C for 2 min (×1), then 95 °C for 10 min (×1), and 95 °C for 15 s and 60 °C for 1 min (×40). For the pig, *Has, Hyal,* and *Gusb* (PrimePCR Probe Assays, BioRad) reactions were incubated at 95 °C for 2 min (×1), then 95 °C for 5 s and 60 °C for 30 s (×40). For cow and pig, *Col1a1, Col3a1,* and *Gusb* (PrimePCR SYBR Green Assays, BioRad) reactions were incubated at 95 °C for 2 min (×1), then 95 °C for 5 s, 60 °C for 30 s (×40), followed by a melt curve analysis. Finally, for mouse, *Col1a1, Col3a1,* and *Gusb* (PrimerBank primer sequences, primers synthesized by Integrated DNA Technologies) reactions were incubated at 95 °C for 10 min (×1), then 95 °C for 15 s, 60 °C for 30 s, and 72 °C for 30 s (×40), followed by a melt curve analysis.

For all RT-PCR assay types, transcript data were normalized to *Gusb* for the same sample, and no template reactions were utilized as negative controls, primer quality was assessed by ensuring <0.5 Ct (threshold cycle) between replicates, and, for SYBR green chemistry, a single amplicon was confirmed after melt curve analysis. Where sequence information is available, we confirmed that the primers amplified cDNA for the species-specific transcript of interest. Fold change values were determined using the 2^−∆∆Ct^ method, and the fold change was expressed over *Has1*, *Hyal1*, or *Col1a1* as appropriate for each graph.

### 2.5. Mouse Kiaa1199 Detection Using RNA In Situ Hybridization

*Kiaa1199* mRNA transcripts were detected in 6–12 week old CB6F1 mice using RNAscope (Advanced Cell Diagnostics, Hayward, CA, USA) in situ mRNA hybridization method as previously described [30]. The probe was synthesized and obtained from the same company, and the RNAscope 2.5 HD Red assay kit was used to amplify and detect the signal. Two ovaries from 2 different mice (1 ovary/mouse, 3 sections/mouse) were hybridized and imaged on an EVOS FL Auto Imaging System using a 20× and 40× objective.

### 2.6. Hyaluronan Binding Protein Assay

To localize hyaluronan (HA) in histologic sections, we performed a fluorescent HA binding protein assay (HABP) as previously described [31], as well as a chromogenic HABP assay. Briefly, bovine (*n* = 2 cows, *n* = 1 ovary/cow, *n* = 4 sections/ovary fluorescent HABP; *n* = 2 cows, *n* = 1 ovary/cow, *n* = 2 sections/ovary chromogenic HABP), porcine (*n* = 2 pigs, *n* = 2 ovaries per pig, *n* = 2 sections/ovary fluorescent HABP; *n* = 2 pigs, *n* = 2 ovaries/pig, *n* = 2 sections/ovary chromogenic HABP), and mouse (*n* = 2 mice, *n* = 1 ovary/mouse, *n* = 4 sections/ovary fluorescent HABP) sections were deparaffinized in CitriSolv and then re-hydrated in a series of decreasing concentrations of ethanol baths (100%, 95%) followed with a wash in water and then PBS, with agitation. An Avidin-Biotin Blocking Kit (Vector Laboratories) was then used to block the endogenous signal. Next, sections were incubated with normal goat serum (1.5% *v/v* in PBS, Vector Laboratories) for 20 min followed by incubation with 1 mg/mL of bovine testis hyaluronidase (Sigma-Aldrich, St. Louis, MO, USA) in a normal saline buffer for 1 h at 37 °C for negative control or saline buffer without hyaluronidase for HA detection. After a rinse in PBS, biotinylated-HABP (1:100 in normal goat serum, Millipore-Sigma) was added to the slides for 1 h at room temperature. Next, the slides were washed with PBS 4 times and incubated with an avidin-biotin complex (Vectastain Elite ABC Kit, Vector Laboratories) for 30 min at room temperature. For fluorescent detection, the signal was further amplified using tyramide signal amplification (TSA Plus Fluorescein System Akoya Biosciences, Marlborough, MA, USA) at 1:400 dilution in reaction buffer for 5 min. Finally, slides were washed 3 times with PBS for 10 min each, mounted with Vectashield containing DAPI (Vector Laboratories), and coverslips affixed with clear nail polish. Images were taken using an EVOS FL Auto Cell Imaging System (ThermoFisher, Waltham, MA, USA) and Leica SP5 (Leica, Wetzlar, Germany). Entire bovine, porcine, and mouse samples were scanned at 10X when imaged with the GFP light cube using the EVOS FL Auto Cell Imaging System. The imaging settings were kept constant for all slides. HABP staining intensity analysis was performed by measuring the HA intensity/area in the total ovarian section using Fiji (ImageJ, National Institute of Health, Bethesda, MD, USA). The background was determined using the hyaluronidase-treated sections, and this value was subtracted from the HA positive sections.

For chromogenic detection, after ABC reagent incubation, samples were incubated with DAB substrate solution (DAB Peroxidase (HRP) Substrate Kit, Vector Laboratories), and color development was followed under the microscope. After 10 min, the reaction was stopped using water. Finally, samples were counterstained with hematoxylin. Briefly, the slides were incubated with hematoxylin for 20 s, washed with running water for 2 min, and then incubated with acid ethanol for 20 s. Next, the slides were washed with running water for 1 min, incubated with bluing reagent for 1 min, and washed again for 1 min with running water. Finally, the slides were dehydrated with serial ethanol washes (80%, 95%, and 100%) for 5 min each and cleared with CitriSolv. HABP chromogenic slides were mounted using Cytoseal and visualized using an EVOS FL Auto Cell Imaging system (tissue scan and single images) and Nikon Eclipse E600 (Nikon, Tokyo, Japan, single images).

### 2.7. Hyaluronan ELISA-like Assay and Molecular Weight Determination in Tissue and Follicular Fluid

Approximately 80 mg of bovine and porcine tissue was added to bead homogenization tubes (Lysing matrix D, MP Biomedical) with 800 µL of RIPA buffer (10 mM NaPyrophosphate,100 mM EDTA, 10% NP40, 10% Na-deoxycholate). The tissues were homogenized in a FastPrep 24 at 6.5 m/s for 60 s and repeated 7 times. After homogenization, the samples were centrifuged at 21,000× *g* for 2 min. Then, the supernatant was removed and immediately placed on ice. Molecular weight cut-off columns (Centristart 1, Sartorius, Stonehouse, UK) were used to separate the HA supernatant into <100 kDa, 100–300 kDa, and >300 kDa molecular weight fractions. HA samples for the different molecular weight fractions were prepared for the HA ELISA-like assay by diluting the samples with RD5–18 buffer per the manufacturer’s instructions. Total HA and HA MW fractions were detected using the HA Quantikine ELISA Kit (R&D Systems, Minneapolis, MN, USA) according to the manufacturer’s instructions. The data were analyzed using a 4-parameter logistic curve (https://myassays.com/four-parameter-logistic-curve.assay, accessed on 14 March 2021).

To remove protein from bovine and porcine follicular fluid samples, 4.5 µL of 1.8 U/mL Proteinase K was added to 50 µL of follicular fluid and incubated at 37 °C in heat blocks for 15 min. After the incubation, the samples were removed from the heat blocks and transferred to Phase Lock Gel Tubes (QuantaBio, Beverly, MA, USA), to which 50 µL of phenol:chloroform:isoamyl alcohol (25:24:1 *v*/*v*/*v*, Fisher Scientific, Waltham, MO, USA) was added and mixed using a laboratory vortex. Samples were centrifuged for 15 min at 14,000× *g* to separate the aqueous phase containing HA from the organic phase. One hundred microliters of phosphate-buffered saline (PBS) was added, mixed by triturating, and then transferred to new Phase Lock Gel Tubes. Then, 50 µL of pure chloroform was added, mixed, and centrifuged for 15 min at 14,000× *g*. The aqueous phase was removed, placed into a 1.5 mL microcentrifuge tube on ice. An additional 50 µL of PBS was added to the sample. Three HA molecular weight pools were created from these follicular fluid samples as described above. HA molecular weight cut-off samples were further prepared by diluting the samples with RD5-18 buffer. HA concentration was measured using a Hyaluronan Quantikine ELISA Kit (R&D Systems) according to the manufacturer’s instructions, and the data were analyzed using a 4-parameter logistic curve (https://myassays.com/four-parameter-logistic-curve.assay, accessed on 14 March 2021).

### 2.8. Statistical Analysis

GraphPad Prism version 8.0.1 was used to plot all graphs and to perform statistical analysis. We used Student’s *t*-test to determine significance between two groups or one-way ANOVA with a Tukey’s post hoc test to determine significance between three or more groups. In both cases, a *p*-value of <0.05 indicates significant differences between groups, and the specific *p*-value is included in each graph where significance was observed, unless that *p*-value was <0.0001, in which case, <0.0001 was used.

## 3. Results

### 3.1. Collagen Localizes to the Follicles, Vasculature, and Stroma of Bovine and Porcine Ovaries

To compare the architecture of bovine and porcine ovaries, H&E staining was performed. H&E staining revealed primordial follicles (quiescent follicles), growing follicles (activated follicles), and antral follicles (pre-ovulatory follicles), the ovarian stroma, and vasculature (Figure 1A,C,E,G). Serial sections were stained with PSR to visualize collagen within the bovine ovary. After PSR staining, collagen appears red and non-collagen-containing structures are labeled a yellowish or faint pink color (Figure 1B) [10,31,32]. Collagen fibers surrounded the primordial follicles (Figure 1D, primordial). In growing follicles, while collagen fibers surround the follicles, the network of collagen fibers immediately surrounding the follicles appeared less dense than at the primordial follicle stage, possibly due to follicle growth and activation (Figure 1D, growing). Antral follicles had intense collagen staining in the theca cell layer surrounding the follicle (Figure 1D antral and antral–inset); there was minimal diffuse PSR-positive staining surrounding the granulosa cells (Figure 1D, antral -inset). Collagen fibers were also prominent in the stroma (Figure 1F) and surrounding the vasculature (Figure 1H). As expected, there was no PSR-positive staining in the oocyte. These findings were validated with Masson’s Trichrome (MTC) staining (Appendix A), another histochemical approach to detect collagen which is stained blue instead of red as seen with PSR staining.

Porcine ovarian tissue sections were also stained with H&E to visualize ovarian tissue architecture (Figure 2A) and specific structures, including follicles (primordial, growing, and antral; Figure 2C), stroma (Figure 2E), and vasculature (Figure 2G). Porcine primordial follicles were associated with a network of collagen fibers surrounding them (Figure 2D, primordial). PSR staining revealed that collagen immediately surrounding growing follicles seemed to form a thicker layer than found around primordial follicles (Figure 2D, primordial vs. growing). Porcine antral follicles had a layer of collagen fibers surrounding the follicle, with a lack of collagen staining within the granulosa layer as observed in the bovine ovary (Figure 2D, antral and antral-inset). Unlike the bovine ovary, we observed dark red pigmentation in some granulosa cells (Figure 2D, antral). However, this was not observed in MTC stained porcine ovarian sections (Appendix A), suggesting this may be a staining artifact. Collagen staining was very robust in the theca cell layer in the porcine ovary relative to the bovine ovary, which may be a species-specific difference (compare Figure 1D and Figure 2D, antral–inset). Collagen fibers were present throughout the porcine ovarian stroma, similar to what was observed in the bovine ovarian stroma (Figure 2F). Collagen fibers were also found in close association with the vasculature (Figure 2H). There was no PSR-positive staining in oocytes of any follicle class. PSR-positive staining in pig ovaries was confirmed with MTC staining (Appendix A).

### 3.2. Bovine Stromal Tissue and Porcine Ovaries Are Enriched in Collagen

To quantify the collagen content in bovine and porcine ovaries that was observed by PSR and MTC staining, we performed a biochemical hydroxyproline assay. Proline hydroxylation is a common post-translational modification in the proline-rich collagen molecule, helping it to twist into its triple helical structure, and thus it can be used as a measure of collagen content [33,34]. We previously reported that ovaries from reproductively young mice contain detectable collagen using this assay, so we used mouse ovaries as a positive control for assay validation and to provide a species comparison [10]. This assay revealed that bovine stromal tissue, porcine ovaries, and whole mouse ovaries have different collagen content (Figure 3A). The average collagen content in cow ovaries was 41 µg/mg of tissue, while pigs had 50% less collagen (20.5 µg/mg of tissue, *p* = 0.0002; Figure 3A). Mouse ovaries had the lowest collagen content, 7 µg/mg of tissue (cow vs. mouse, *p* = 0.0057 and pig vs. mouse, *p* = 0.0202; Figure 3A), and the level of collagen was consistent with what we previously reported [10].

To understand the species differences in collagen content in the ovary and how it is being regulated, we assessed transcripts from fibrillar collagen genes. To determine the relative abundance of type I and type III collagen, we measured *Col1a1* and *Col3a1* transcripts by RT-PCR. In bovine stromal tissue, *Col3a1* was more abundant than *Col1a1*, but this did not reach statistical significance (Figure 3B).In porcine ovaries, we observed the opposite, with a trend for *Col1a1* being more abundant in comparison to *Col3a1* (Figure 3C). Mouse ovaries were used as a positive control for assay validation and to provide a species comparison. In mouse, we observed 16-fold more *Col3a1* than *Col1a1* (Figure 3D; *p* < 0.0001).

### 3.3. Hyaluronan Localizes to the Follicles, Vasculature, and Stroma of Bovine and Porcine Ovaries

Due to the association between collagen and HA in maintaining tissue biomechanics, we assessed the localization of HA in the ovary to observe if collagen and HA localization parallel each other. We localized HA in bovine and porcine ovaries using the HABP assay as previously described [3,9,21]. Whole ovarian sections were H&E stained to observe ovarian architecture (Figure 4A), which includes structures such as follicles (primordial, growing, and antral; Figure 4C), stroma (Figure 4E), and vasculature (Figure 4G). HABP staining of serial sections revealed HA throughout the ovary, and hyaluronidase pre-treatment abolished the HA signal, confirming the assay specificity (Figure 4B and inset). In bovine ovaries, HA was found immediately surrounding primordial follicles and growing follicles (Figure 4D, primordial and growing). At the antral follicle stage, HA was found in the follicular fluid and between individual cumulus and theca cells (Figure 4D, antral and antral-inset). The bovine ovarian stroma is enriched in HA (Figure 4F), and intense HA staining was found surrounding the vasculature (Figure 4H). These findings were validated with HABP chromogenic staining (Appendix A), in which positive HA signal is brown instead of green.

To evaluate HA localization in porcine ovaries, tissue sections were stained with H&E to reveal ovarian architecture (Figure 5A,C,E,G), while serial tissue sections were used in the HABP assay. Like bovine ovaries, porcine ovaries were HA-rich structures. HA was distributed around primordial follicles and growing follicles (Figure 5D, primordial and growing). In porcine antral follicles, HA was found between the granulosa cells and appeared more than what was found between the granulosa cells of the bovine antral follicle (Figure 5D, antral). Moreover, there was robust HA staining in the theca cell layer (Figure 5D, antral and inset). HA was distributed in the porcine antral follicle in a graded pattern with more HA found around the cumulus cells followed by decreased HA within the granulosa cell layer, then increased HA in the theca cell layer (Figure 5D, antral); however, further analysis is needed to determine if this was consistent among antral follicles. Finally, HA was prominent in the stroma (Figure 5F) and encircled the vasculature (Figure 5H). These findings were validated with HABP chromogenic staining (Appendix A), in which positive HA signal is brown instead of green.

Next, we compared ovarian HA content in cow, pig, and mouse in parallel to determine if there are species-specific differences in HA content and localization, with mouse ovaries serving as a control. We quantified the staining intensity per area of tissue and found that porcine ovaries had the most HA (compare Figure 6A vs. Figure 6B vs. Figure 6C). Although differences in ovarian HA content did not achieve statistical significance, there appeared to be 20% more HA in porcine ovaries when compared to bovine ovaries, 36% more HA in porcine ovaries when compared to mouse ovaries, and 15% more HA in bovine ovaries when compared to mouse ovaries (Figure 6D).

### 3.4. Ovarian Tissue and Follicular Fluid HA Content and Molecular Weight Distribution Is Species-Specific

HA biological activity is determined by its molecular weight, and HA can exist in a spectrum of polymer sizes from HMW (≥1000 kDa) to LMW (≤250 kDa) [17]. HMW HA is associated with tissue homeostasis, while LMW HA are generated when tissues are injured and is associated with inflammation [17]. Thus, given the importance of HA MW distribution to its physiologic function, we isolated and measured total HA and then used MW cut-off columns to fractionate bovine stromal tissue HA and porcine ovarian HA into three different MW ranges <100, 100–300, and >300 kDa. Then, we quantified the HA amount in each of these size pools using an ELISA-like assay. There was more total HA in porcine ovaries (5.4 ng/mg of tissue) than bovine ovaries (0.15 ng/mg tissue; Figure 7A; *p* = 0.0414), which is consistent with what we observed with the HABP staining (Figure 6). Moreover, porcine ovaries tended to have more HA than bovine ovaries for each MW distribution assessed. Most of the HA in the bovine and porcine tissue tended to exist in the >300 kDa fraction, with bovine stromal tissue having an average of 0.37 ng/mg of tissue of HA while porcine ovaries had an average of 7.51 ng/mg of tissue (Figure 7B). For the 100–300 kDa fraction, bovine stromal tissue had an average of 0.03 ng/mg of tissue of HA, while porcine ovaries had an average of 0.26 ng/mg of tissue of HA (Figure 7B). For the <100 kDa fraction, bovine stromal tissue had 0.01 ng/mg of tissue of HA, while porcine ovaries had less HA, 0.01 ng/mg of tissue (Figure 7B). The observation that most HA appeared to exist in the >300 kDa fraction for bovine stromal tissue and porcine ovaries is suggestive that normal tissue homeostasis is being maintained.

Another critical microenvironment in the ovary is the follicular fluid found within antral follicles. The follicular fluid is in direct contact with the cumulus-oocyte complex, and thus, changes in follicular fluid HA content or HA fragment distribution could have important impacts on cumulus-oocyte complex biology. Surprisingly, in the follicular fluid, we observed the inverse of what was found in the tissues, with a trend to overall more HA in the follicular fluid of bovine ovaries relative to porcine ovaries, but no clear trends between HA content in various size pools were noted (Figure 7C).

### 3.5. Predominant Ovarian Hyaluronan Synthase and Hyaluronidase Transcript Content Differs between Cows and Pigs

HA content in tissues is regulated by HA production by HA synthases and degradation by hyaluronidases and ROS. To determine which HA synthases and hyaluronidases were expressed in bovine stromal tissue and porcine ovarian tissue, we profiled *Has1*, *Has2*, and *Has3* (HA synthases) and *Hyal1*, *Hyal2*, *Tmem2*, and *Kiaa1199* (hyaluronidases) using real-time PCR. Assessing HA synthases in mouse as a control, we observed that all three HA synthases were expressed with *Has3* being the most abundant transcript, followed by *Has2,* which was approximately 2.6-fold less than *Has3* (Appendix A; *p* < 0.0001, *Has1* vs. *Has2* and *Has1* vs. *Has3*). This paralleled our previous finding of HA synthase transcripts in mice [9]. All four hyaluronidase transcripts were expressed in mouse ovaries, with *Kiaa1199* being the most abundant transcript (Appendix A; *p* < 0.0001 *Hyal1* vs. *Hyal2*, *Hyal1* vs. *Tmem2*, *Hyal1* vs. *Kiaa1199*). Relative to *Hyal1*, *Hyal2* was 4-fold less, *Tmem2* was 1-fold less than *Hyal1,* and *Kiaa1199* was 20-fold more than *Hyal1* (Appendix A), again matching our previously published data [9]. *Has1*, *Has2,* and *Has3* transcripts were all expressed in bovine ovarian tissue. *Has3* tended to be the most abundant synthase and was approximately 5-fold more than *Has1*, and *Has2* tended to be the second most abundant transcript having approximately 3.7-fold more than *Has1* (Figure 8A). The hyaluronidases *Hyal1*, *Hyal2*, *Tmem2,* and *Kiaa1199* were assessed in bovine stromal tissue, and all four of the hyaluronidase transcripts were found; *Hyal2* transcripts appeared most abundant, *Kiaa1199* transcripts appeared the least abundant, and there was a significant difference between these two bovine hyaluronidases (Figure 8B, *p* = 0.0204). All three HA synthase transcripts were found in porcine ovaries; however, we observed the inverse from that observed in bovine ovaries. In porcine ovaries, *Has2* transcripts appeared most abundant, and there was an approximately 5.4-fold difference between *Has2* and *Has1* (Figure 8C). While *Has3* transcripts were the most abundant in bovine stromal tissue, they appeared the least abundant in porcine ovaries (Figure 8C). In porcine ovaries, all hyaluronidase transcripts were expressed, and *Hyal1* and *Hyal2* trended towards least abundant and had similar levels, while *Tmem2* and *Kiaa1199* trended towards most abundant transcripts (Figure 8D). As in the mouse, porcine ovaries abundantly expressed *Kiaa1199* while in bovine stromal tissue, *Hyal2* was the most abundant and *Kiaa1199* was the least abundant hyaluronidase suggesting either species-specific differences or ovarian compartment differences. For example, using RNAscope technology, we localized *Kiaa1199* transcripts within mouse ovaries and observed prominent *Kiaa1199* expression in the corpora lutea, suggesting a very compartment-specific function (Appendix A).

## 4. Discussion

In this study, we used qualitative and quantitative approaches to characterize collagen and HA content within bovine and porcine ovaries. Collagen and HA are prominent throughout the ovaries of these larger mammalian species and localize within specific ovarian compartments, which may indicate different physiological roles for these two matrix molecules. Due to the close interaction of the stroma with the follicle and the potential impact of the stroma on follicle quality, previous studies were conducted using bovine and porcine ovaries to characterize this critical ovarian compartment [23,24,25,26,27,28,29]. Our findings of HA localization in bovine ovaries are consistent with a previous study reporting the localization of HA in the stroma, theca cell layer, and the COC within the bovine ovary [30]. They also reported minimal HA localization in the corpora lutea, which we did not observe because we selected against ovaries with corpora lutea [30]. Another study assessed matrisome proteins in decellularized porcine ovaries using a proteomic approach and found that 82 core matrisome proteins such as collagens and proteoglycans were differentially expressed across different depths of the ovary [26]. Based on the methods employed, that study precluded an analysis of HA. This is because HA is a glycosaminoglycan that lacks a core protein found in proteoglycans. Our manuscript extends these published studies by focusing on two ovarian stromal ECM molecules in detail and by directly comparing collagen and HA content between two large mammalian species. Despite the low number of biological replicates between species, this work is important because the ovarian stroma, in which collagen and HA are found, is a microenvironment which could have significant impacts on follicle growth and oocyte quality in health, disease, and aging.

Collagen plays a role in various physiological processes such as development and ovulation. Furthermore, ovarian collagen is dynamically regulated postnatally as well. For example, ovarian collagen is higher in young girls (0–10 years old) than in those 11–20 years old; collagen increases again in women 51 years old and older [9]. Collagen content is reduced during ovulation in species such as rats [35], rabbits [36], hamsters [37], sheep [38], and women [39], illustrating the cycle-specific dynamic nature of ovarian collagen content. A caveat of our study is that we did not assess the cycle-specific dynamic nature of collagen in these bovine and porcine ovaries to be able to compare it to what was previously observed in other species. However, using PSR and MTC staining of bovine and porcine ovaries, we observed compartment-specific collagen localization in various ovarian sub-compartments such as in the follicles which have prominent collagen in the theca cell layer, but no staining in the granulosa cell layer. The theca layer is divided into the theca interna and theca externa. The theca interna layer is involved in the production of androstenedione, while the theca externa layer has macrophages, fibroblasts and smooth muscle cells which provide structural support [40]. Collagen I and III are present in the theca externa layer of rat follicles [41]. In humans, collagen I is also present in the theca externa cell layer, but not in the theca interna or the granulosa cell layer [42]. Collagen I is present in the theca cell layer and the stroma of bovine ovaries [43]. Similarly, we detected collagen I and III in bovine and porcine ovaries. A caveat of our study is that only histochemical staining and a qualitative staining analysis was performed. Analysis of fiber orientation, thickness, length, and localization may be informative and reveal important species-specific and/or functional differences across the different ovarian compartments and is a subject for future studies. However, we did make use of the hydroxyproline assay to estimate ovarian collagen content quantitatively. We found more collagen in bovine ovarian tissue when compared to porcine ovaries. This may be because follicles were removed from the bovine ovarian tissue creating a tissue sample enriched in stroma in contrast to the porcine ovarian tissue which was left intact. Future studies will utilize porcine ovaries after follicle removal which will provide a better comparison between species.

In addition to collagen, HA is another important and prominent ovarian ECM molecule found in the mammalian ovary [9]. Pig ovaries had the most HA, followed by cow and mouse ovaries, and, perhaps due to the fewer numbers of larger follicles in our cow and pig samples, HA staining appeared more homogeneous than what we observed in mice. In mice, there is a dramatic increase in HA in secondary follicles relative to primordial/primary follicles, which was difficult to appreciate in the cow and pig ovaries [9]. Total ovarian HA decreases with age in mice and in humans, and this occurs in the stromal compartment and in the theca cell layer of the mouse ovary with reproductive age [9]. Whether this observation is conserved in cows and pigs needs to be evaluated. HA MW determines its biological activity, and HA can range from HMW (≥1000 kDa) to LMW (≤250 kDa), with HMW HA creating a homeostatic ECM, while LMW HA creates a reactive ECM [17]. Consistently, recent work from our group demonstrated that LMW HA (200 kDa) induced a Th2 cytokine profile in ovarian stromal cells [21]. Most HA in bovine and porcine tissue existed in the >300 kDa fraction, suggesting a homeostatic ECM, but this may change with advanced reproductive age.

It is interesting to consider ovulatory differences between cows and pigs and how ovarian structure and matrix composition may differ between these two species. Our data revealed that porcine ovaries contain more HA than bovine ovaries, and this was supported by the HABP assay (whole ovary tissue sections) and the quantitative HA ELISA-like assay. Moreover, porcine ovaries have less collagen than bovine ovaries, as determined using the hydroxyproline assay from which collagen content can be estimated. Given that the pig is polyovulatory and the cow mono-ovulatory, this suggests that increased ovarian HA and reduced ovarian collagen relative to cows are important to support the simultaneous development of multiple follicles in pigs. Future studies should examine matrix structural and compositional differences in additional mono-ovulatory and polyovulatory species to determine if this finding is penetrant and identify other ovarian parameters that correlate with ovulatory status to expand upon this intriguing concept.

HA and proteoglycans help in the formation of follicular fluid by creating an osmotic gradient that recruits plasma from capillaries in nearby theca cells [44]. The follicular fluid in human, cow, and pig ovaries makes up a large portion of the follicles at ovulation and is a microenvironment that could affect oocyte quality [45]. One study shows bovine follicular fluid HA size ranges from 400 kDa to 2 MDa [44]. Our findings showed that there tended to be more HA in bovine follicular fluid relative to porcine follicular fluid and bovine follicular fluid, but due to variability between the two samples in each species, we cannot draw conclusions regarding HA size pools at this time which is certainly an important subject for future research. Conditions such as aging alter follicular fluid composition and affect oocyte quality in cows [46,47]. We noticed that HA MW distribution in the tissues and follicular fluids of pigs were similar, possibly due to the presence of follicles within the tissues, while in cows, the HA MW distribution was more different in the follicular fluid and the stroma enriched tissue. A caveat of this study is that even though we know the life expectancy for these animals, and their ages suggest that they are reproductively young, limited information is available on their fertility, and we have not extrapolated the ages of the bovine and porcine ovaries to each other or to human ovaries. Thus, although we assume that the primary differences in HA are species-specific, they may be attributed to differences in age. Future work is needed to dissect the role of HA in fertility and aging in cows and pigs and determine if these parameters parallel what is found in humans.

To assess how HA is regulated in bovine and porcine ovaries, we measured HA synthase and hyaluronidase transcript levels. In bovine stromal tissue and in mouse ovarian stroma, *Has3* was most abundant, while in porcine ovaries, *Has2* tended to be most abundant. The difference in *Has3* transcript abundance in mice and cows relative to pigs may reflect the presence of oocytes in the pig ovary, which is lacking in the murine and bovine tissue used for transcript analysis. Our data suggest that, in the mouse and cow, Has3 may be predominantly involved in stromal HA synthesis. However, further research should be conducted using only stroma from pig ovaries which will allow us to directly compare between species for a better understanding of compartment-specific gene expression. There also appeared to be species differences observed for the predominant hyaluronidase. In mouse ovarian stroma, *Kiaa1199* was the most predominant hyaluronidase transcript while in porcine ovaries, *Kiaa1199* and *Tmem2* appeared most abundant, and, in bovine ovarian stromal tissue, *Hyal2* appeared most abundant, and *Kiaa1199* appeared least abundant. *Kiaa1199* being the least expressed hyaluronidase in bovine ovaries, could be due to the lack of the corpora lutea in the bovine tissue we utilized. Indeed, using RNAscope technology, we found that *Kiaa1199* exhibits compartment-specific localization within mouse corpora lutea suggesting that *Kiaa1199* regulates how much HA is present in this specific ovarian compartment. Ongoing studies are profiling the location of *Has* and *Hyal* transcripts within mammalian ovaries to determine where each gene is expressed and from which hypotheses regarding specific roles for each enzyme in ovarian biology can be proposed and tested.

Although we had a small number of biological replicates in each species, these data nevertheless have important scientific value. Indeed, we were surprised to see that, despite the lack of statistical differences, the biological replicates we achieved were oftentimes close to one another and did not overlap with other experimental groups in the same assay. This is more impressive when one considers the many possible sources of variability including, age, estrous cycle stage, and ovulatory status between species, and demonstrates a robust phenotype despite lacking this information. Finally, where orthogonal assays were employed, both assays supported the same conclusion. For example, while HABP staining suggested that the porcine ovary contains more HA than the cow ovary, the quantitative HA ELISA-like assay confirmed this finding and was supported by statistical significance. Thus, we are confident that this body of work contributes significantly to our understanding of mammalian ovarian matrix biology.

It is clear from the present study and other studies on the ovarian stroma [3,9,10,24,25,27,28,29] that characterizing this microenvironment in health, aging, and disease is of critical importance. For example, the role of the ovarian stroma is best understood in polycystic ovary syndrome (PCOS), in which the stroma is expanded and accompanied by increased collagen accumulation in the tunica albuginea and the ovarian cortex [48]. Moreover, women with PCOS also have increased ovarian stiffness in comparison to their aged-match controls, suggesting a micromechanical consequence of increased collagen in PCOS ovaries [49]. There is also an increase in androgen synthesis by theca cells in PCOS patients resulting in amenorrhea, infertility, and hirsutism, and this is presumably due to increased thickness of the theca cell layer [50,51]. Thus, PCOS patients provide evidence demonstrating the significant negative impacts of excessive ovarian stroma and the importance of characterizing this critical ovarian subcompartment in mammals. Altogether, this study broadens our understanding of ovarian stromal biology in mammalian species with human-like reproductive parameters and provides a foundation for further exploration of ovarian microenvironments in women and in critical agricultural species.

## 5. Conclusions

Using molecular and histochemical approaches we observed that HA and collagen are prominent extracellular matrix molecules in bovine and porcine ovaries. These ovaries exhibit species specific differences in the localization and quantity of HA and collagen as well as the expression of HA- and collagen- related genes, which may be important for creating and maintaining proper follicle development in these monovulatory and polyovulatory animals.

## Figures and Tables

**Figure 1 genes-12-01186-f001:**
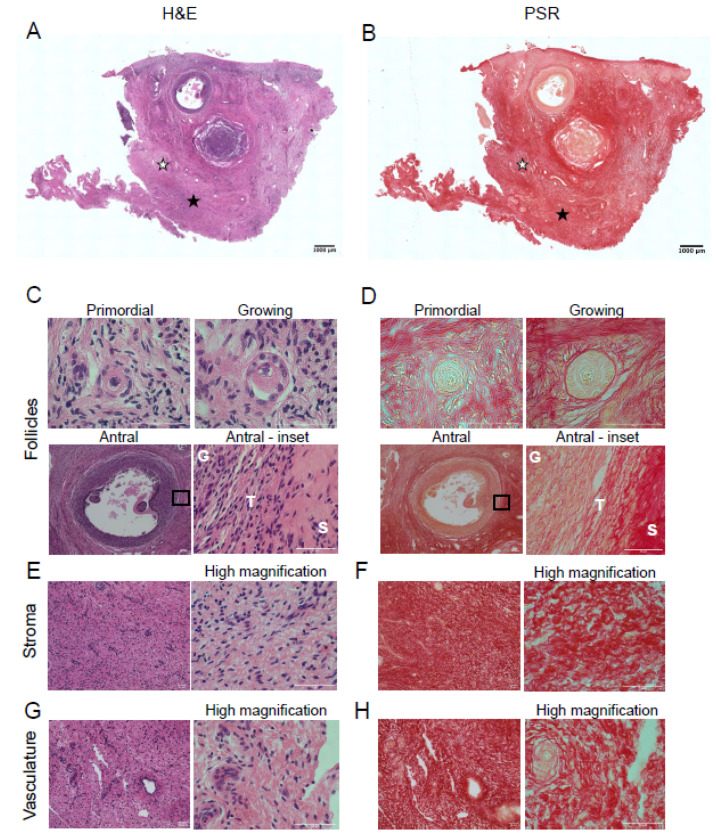
Picrosirius red staining detects collagen in bovine ovarian follicles, stroma, and vasculature. Hematoxylin and eosin (H&E) staining (**left**) and picrosirius red (PSR) staining of bovine ovarian tissue sections (**right**). (**A**) H&E staining illustrates the microanatomical structure of a bovine ovarian tissue section. (**B**) PSR staining localizes collagen (red) in a section sequential to that of the one seen in (**A**). H&E and PSR staining in (**C**,**D**) primordial follicle, growing follicle, and antral follicle, (**E**,**F**) stroma, (**G**,**H**) vasculature, respectively. The area demarcated by the black rectangles in low magnification images (**C**,**D**—Antral follicle) is shown in higher magnification images to the right. High magnification images of stroma (**E**,**F**) and vasculature (**G**,**H**) are seen to the right of each low magnification image. G—granulosa cell layer, T—theca cell layer, S—stroma. The scale bars in (**A**,**B**) are 1000 µm and in (**C**–**H**) are 50 µm. Images are representative of bovine ovaries (*n* = 2 cows, *n* = 1 ovary/cow, *n* = 2 sections/ovary). White stars indicate location of vasculature, while black stars indicate location of the stroma.

**Figure 2 genes-12-01186-f002:**
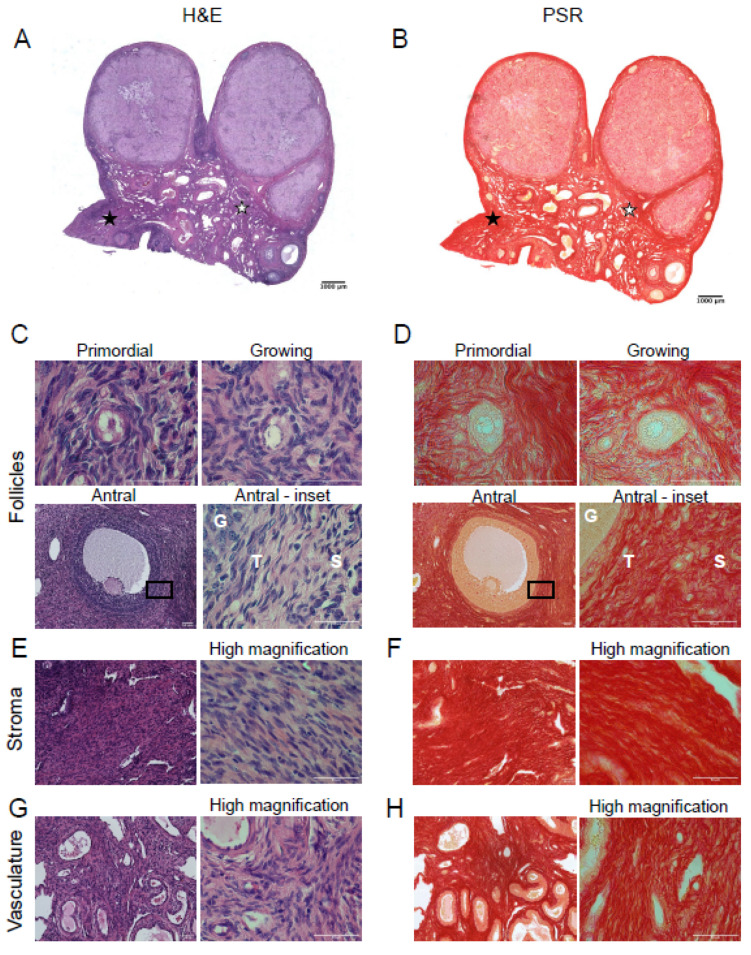
Picrosirius red staining detects collagen in porcine ovarian follicles, stroma, and vasculature. Hematoxylin and eosin (H&E) staining (**left**) and picrosirius red (PSR) staining of porcine ovarian tissue sections (**right**). (**A**) H&E staining illustrates the microanatomical structure of a porcine ovarian tissue section. (**B**) PSR staining localizes collagen (red) in a section sequential to that of the one seen in (**A**). H&E and PSR staining in (**C**,**D**) primordial follicle, growing follicle, and antral follicle (note that the antral follicle image was taken from a different porcine tissue section than shown in (**A**,**B**), (**E**,**F**) stroma, (**G**,**H**) vasculature, respectively. The area demarcated by the black rectangles in low magnification images (**C**,**D**—antral) is shown in higher magnification images to the right. High magnification images of stroma (**E**,**F**) and vasculature (**G**,**H**) are seen to the right of each low magnification image. G—granulosa cell layer, T—theca cell layer, S—stroma. The scale bars in (**A**,**B**) are 1000 µm and in (**C**–**H**) are 50 µm. Images are representative of porcine ovaries (*n* = 2 pigs, *n* = 2 ovaries/pig, *n* = 2 sections/ovary). White stars indicate location of vasculature, while black stars indicate location of the stroma.

**Figure 3 genes-12-01186-f003:**
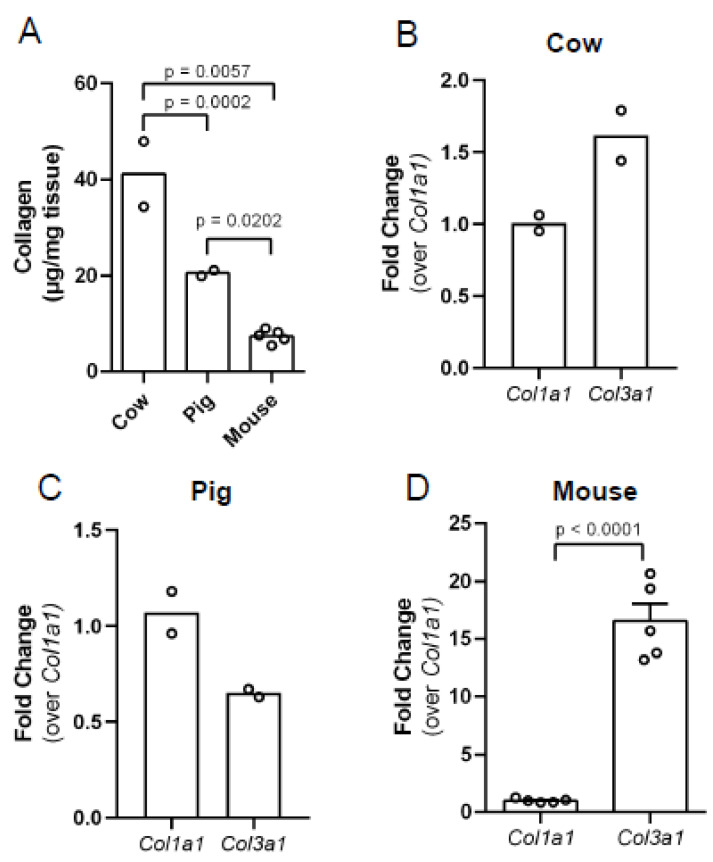
Collagen protein and type I collagen transcripts in bovine ovarian stromal tissue and whole porcine and mouse ovaries. (**A**) Collagen content in bovine ovarian stromal tissue, porcine ovaries, and mouse ovaries determined by the hydroxyproline assay. Relative gene expression of the collagen genes *Col1a1* and *Col3a1* in (**B**) bovine ovarian stromal tissue, (**C**) porcine ovaries, (**D**) mouse ovaries. The data are shown as a fold change in transcript content over *Col1a1*. *n* = 2 cow ovaries, *n* = 2 pig ovaries, *n* = 5 mouse ovaries. Circles represent values from individual animals. Statistical analysis was performed using the Student’s t-test. *p*-values are as indicated.

**Figure 4 genes-12-01186-f004:**
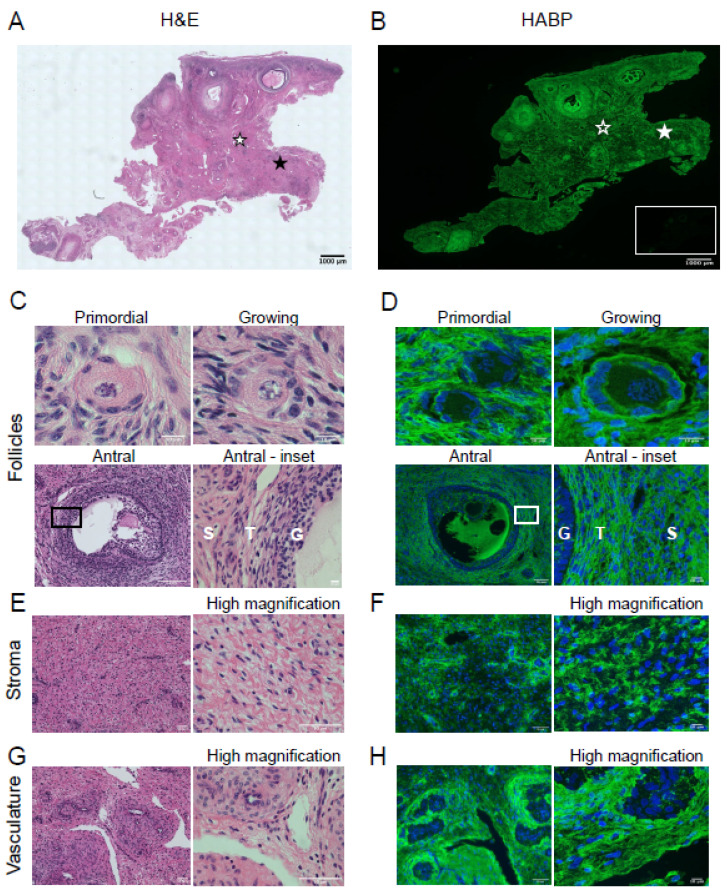
Bovine ovaries are hyaluronan rich. H&E staining (**left**) and HA localization (**right**) in bovine ovarian tissue sections. (**A**) H&E staining showing the microanatomical structures in the bovine ovary. (**B**) HABP staining localizing HA (green) in the bovine ovary. The inset illustrates HABP staining after pretreating a serial ovarian tissue section with hyaluronidase. (**A**) H&E and HABP staining in (**C**,**D**) primordial follicle, growing follicle, and antral follicle, (**E**,**F**) stroma, (**G**,**H**) vasculature, respectively. The area demarcated by the black rectangles in low magnification images (**C**,**D**—Antral follicle) is shown in higher magnification images to the right. High magnification images of stroma (**E**,**F**) and vasculature (**G**,**H**) are seen to the right of each low magnification image. For H&E images, black stars indicate location of the stroma, and white stars indicate location of the vasculature (**A**). For HABP images, the solid white star indicates location of the stroma and the star with the white outline indicates location of the vasculature (**B**). G—granulosa cell layer, T—theca cell layer, S—stroma. The scale bars are (**A**,**B**) 1000 µm and (**C**,**D**) primordial follicle 10 µm, antral follicle 50 µm, antral-inset 10 µm. (**E**,**G**) 50 µm, (**F**,**H**) HABP staining 50 µm, high magnification 10 µm. Images are representative of bovine ovaries (*n* = 2 cows, *n* = 1 ovary/cow, *n* = 4 sections/ovary).

**Figure 5 genes-12-01186-f005:**
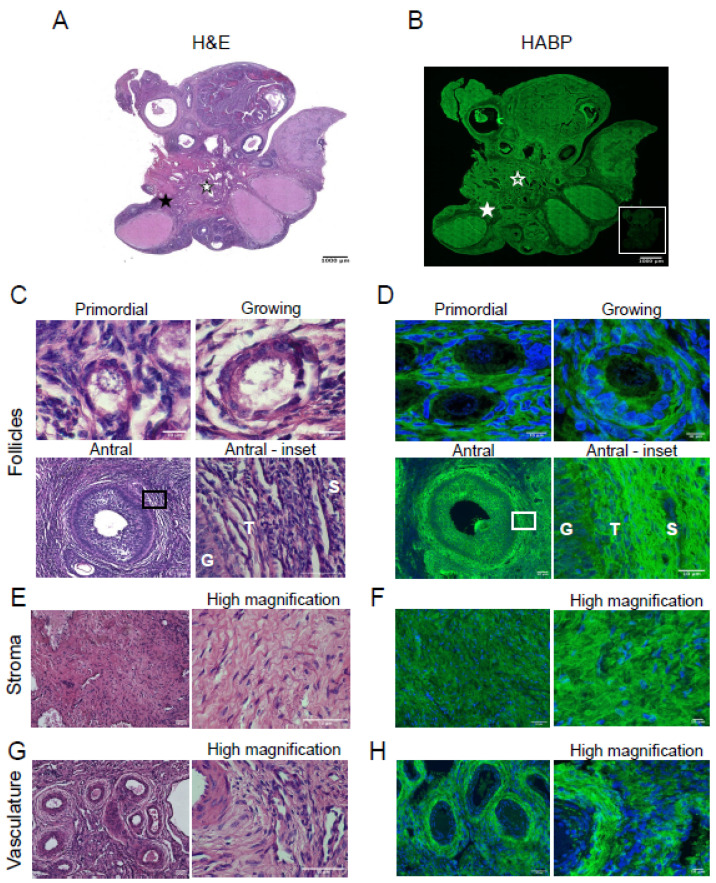
Porcine ovaries are hyaluronan rich. H&E staining (**left**) and HA localization (**right**) in porcine ovarian tissue sections. (**A**) H&E staining showing the microanatomical structures in the porcine ovary. (**B**) HABP staining localizing HA (green) in the porcine ovary. The inset illustrates HABP staining after pretreating a serial ovarian tissue section with hyaluronidase. H&E and HABP staining in (**C**,**D**) primordial follicle, growing follicle, and antral follicle, (**E**,**F**) stroma, (**G**,**H**) vasculature, respectively. The area demarcated by the black rectangles in low magnification images (**C**,**D**—Antral follicle) is shown in higher magnification images to the right. High magnification images of stroma (**E**,**F**) and vasculature (**G**,**H**) are seen to the right of each low magnification image. For H&E images, black stars indicate location of the stroma, and white stars indicate the location of vasculature (**A**). For HABP images solid white stars represents stroma location and white star outline represent the vasculature (**B**). G—granulosa cell layer, T—theca cell layer, S—stroma. The scale bars are (**A**,**B**) 1000 µm and in (**C**,**D**) primordial follicle 10 µm, antral follicle(50 µm, antral-inset 10 µm. (**E**,**G**) 50 µm, (**F**,**H**) HABP staining 50 µm, high magnification 10 µm. Images are representative of porcine ovaries (*n* = 2 pigs, *n* = 2 ovaries/pig, *n* = 2 sections/ovary).

**Figure 6 genes-12-01186-f006:**
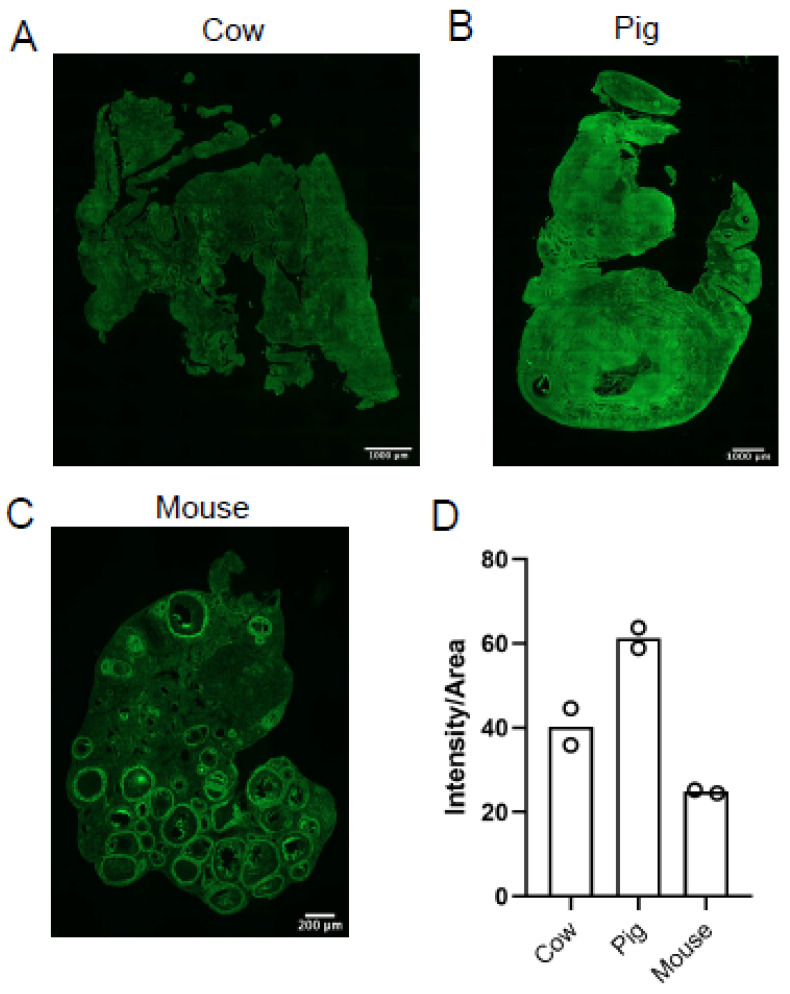
Hyaluronan content in cow, pig, and mouse ovaries. HABP assay was used to detect HA (green) in (**A**) cow, (**B**) pig, and (**C**) mouse ovaries. (**D**) Graph showing HA intensity per area in cow, pig, and mouse ovaries. Circles represent values from individual animals. The scale bars are 1000 µm (**A**,**B**) and 200 µm (**C**). Images are representative of bovine (*n* = 2 cows, *n* = 1 ovary/cow, *n* = 4 sections/ovary) porcine (*n* = 2 pigs, *n* = 2 ovaries/pig, *n* = 2 sections/ovary) and mouse (*n* = 2 mice, *n* = 1 ovary/mouse, *n* = 4 sections/ovary) ovaries.

**Figure 7 genes-12-01186-f007:**
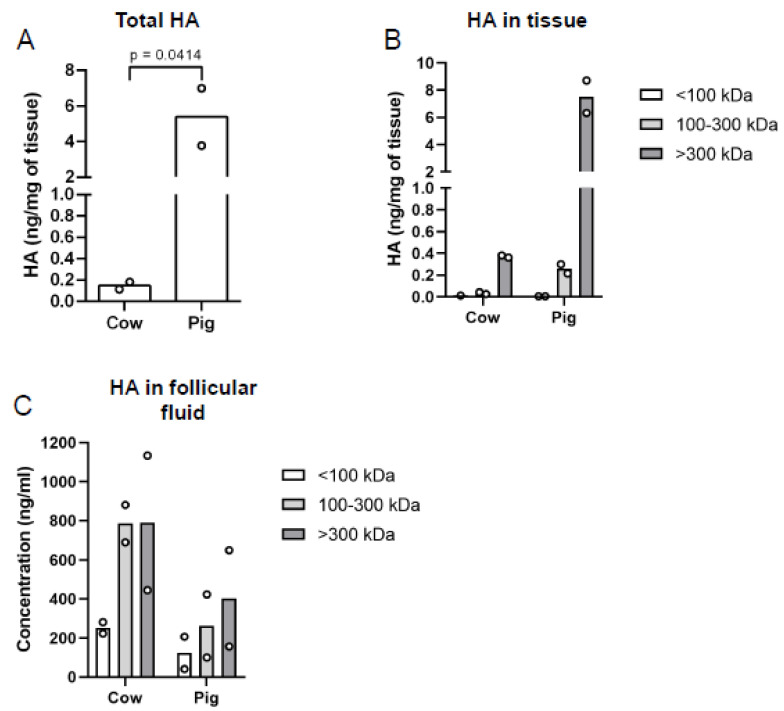
Bovine and porcine total and molecular weight fractionated HA in ovarian tissue and follicular fluid. (**A**) Total HA content in bovine and porcine ovarian tissue was measured using an ELISA–like assay. Column fractionation was used to separate bovine and porcine ovarian tissue and follicular fluid HA into three molecular weight fractions, <100 kDa, 100–300 kDa, and 300 kDa, and then HA concentration in each fraction was measured using an HA ELISA-like assay for (**B**) tissue and (**C**) follicular fluid. Circles represent values from individual animals. The *p*-value is as indicated.

**Figure 8 genes-12-01186-f008:**
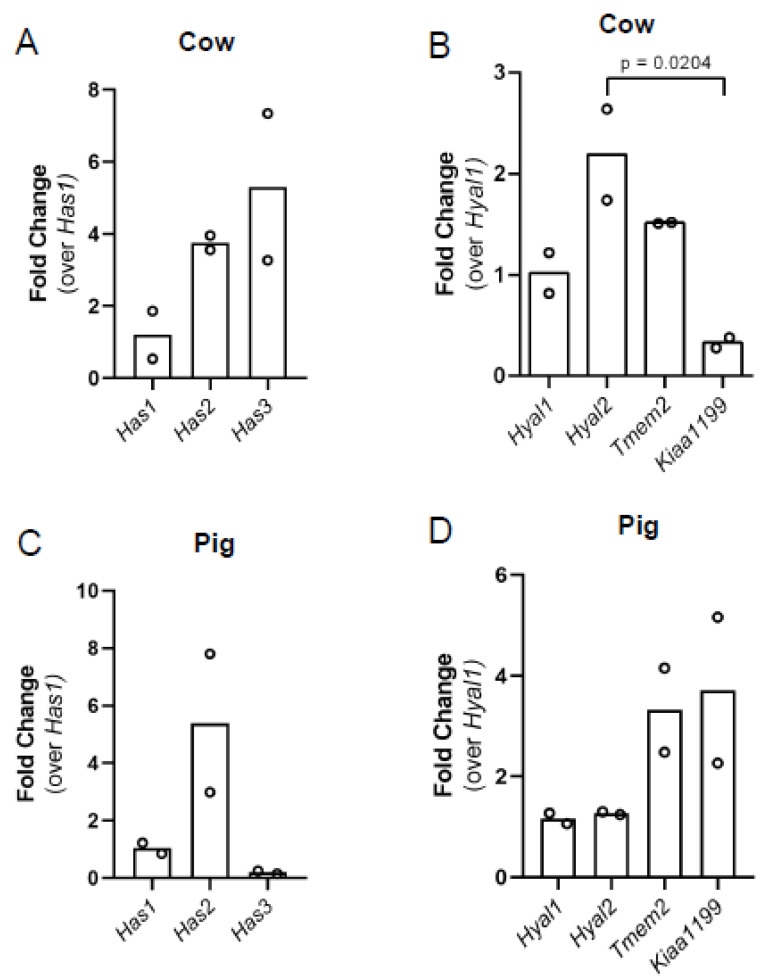
Bovine ovarian stroma and porcine ovaries hyaluronan synthase and hyaluronidase transcript content. Relative hyaluronan synthase gene expression in (**A**) cows and (**C**) pigs. Relative hyaluronidase gene expression in (**B**) cows and (**D**) pigs. Hyaluronan synthase transcript data are shown as fold change over *Has1*, while hyaluronidase transcript data are shown as a fold change over *Hyal1*. *n* = 2 cow ovaries, *n* = 2 pig ovaries. Circles represent values from individual animals. Statistical analysis was performed using a one-way ANOVA. The *p*-value is as indicated.

## Data Availability

Data are available from the corresponding author upon reasonable request.

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
