# Peer review of "Hyaluronan and Collagen Are Prominent Extracellular Matrix Components in Bovine and Porcine Ovaries"

_genes, 2021, doi:10.3390/genes12081186_

Round 1

Reviewer 1 Report

Methodology-qPCR

What is the qPCR thermocycler conditions used?

Where there any PCR controls (Positive or negative controls, reverse transcriptase control?

Line 313-314: Provide more details on statistical analysis. Statistical significance declare at .05 or .01?

Reviewer 2 Report

The Authors in this study entitled “Hyaluronan and Collagen are Prominent Extracellular Matrix Components in Bovine and Porcine Ovaries” described the localization and distribution of collagen and hyaluronan in bovine and porcine ovaries both at the morphological and molecular levels. The paper is essentially descriptive based on two very common ECM components, namely collagen and hyaluronan and the enzymes for synthesis/degradation of hyaluronan.

The Authors’ labs have already used and developed the experimental procedures in previous works in mice where the characteristics of the matrix have been described in relation to aging (references #5, #6 and #7) and to changes that age implies on the characteristics of the components already mentioned, above all hyaluronan (reference #18).

The limit of this work is that it is essentially descriptive and, furthermore, it reaches conclusions on the basis of a too small number of data, being able to describe only trends (Figs 3B, 3C; 6D; 7A, 7B, 7C; 8A, 8C, 8D) or, in some cases, minimal statistical significances (p <0.05) based on only two data per sample (Fig 3A and 8B). Then the samples/data should be increased in number, otherwise the Results are most of the time overestimated because describe a "trend" that does not have scientific value.

Moreover, in the Introduction, the Authors do not sufficiently describe and take into account the fact that the ovary is an highly dynamic tissue undergoing remodelling due to structural and functional changes that occur continuously along the reproductive life and not only related to age-dependent events - that they have been studying extensively in mouse. Also for this reason, the differences between bovine and porcine ovaries described in the manuscript and relied on few data (one ovary/animal on two animals per species) are even more unreliable. Furthermore, it should be taken into account and recall also the fact that cows and pigs are different in the numbers of ovulated oocytes, being monovular and polyovular species, respectively, and this should make the ovarian structure quite different per se.

The limitation of the study is quite clear to the Authors as well, since in the Discussion section, they described three caveat (lines 571-573, 583-584 and 622-26) related to their analyses.
